### **GRACILE:** A comprehensive climatology of atmospheric gravity wave parameters based on satellite limb soundings

Manfred Ern<sup>1</sup>, Quang Thai Trinh<sup>1</sup>, Peter Preusse<sup>1</sup>, John C. Gille<sup>2,3</sup>, Martin G. Mlynczak<sup>4</sup>, James M. Russell III<sup>5</sup>, and Martin Riese<sup>1</sup>

<sup>1</sup>Institut für Energie- und Klimaforschung – Stratosphäre (IEK–7), Forschungszentrum Jülich GmbH, 52425 Jülich, Germany <sup>2</sup>Center for Limb Atmospheric Sounding, University of Colorado at Boulder, Boulder, Colorado, USA. <sup>3</sup>National Center for Atmospheric Research, Boulder, Colorado, USA.

<sup>4</sup>NASA Langley Research Center, Hampton, Virginia, USA.

<sup>5</sup>Center for Atmospheric Sciences, Hampton University, Hampton, Virginia, USA.

Correspondence to: M. Ern (m.ern@fz-juelich.de)

**Abstract.** Gravity waves are one of the main drivers of atmospheric dynamics. The spatial resolution of most global atmospheric models, however, is too coarse to properly resolve the small scales of gravity waves, which range from tens to a few thousand kilometers horizontally, and from below 1 km to tens of kilometers vertically. Gravity wave source processes involve even smaller scales. Therefore, general circulation models (GCMs) and chemistry climate models (CCMs) usually parametrize

the effect of gravity waves on the global circulation. These parametrizations are very simplified. For this reason, comparisons with global observations of gravity waves are needed for an improvement of parametrizations and an alleviation of model biases.

We present a gravity wave climatology based on atmospheric infrared limb emissions observed by satellite (GRACILE). GRACILE is a global data set of gravity wave distributions observed in the stratosphere and the mesosphere by the infrared

- limb sounding satellite instruments High Resolution Dynamics Limb Sounder (HIRDLS) and Sounding of the Atmosphere using Broadband Emission Radiometry (SABER). Typical distributions (zonal averages and global maps) of gravity wave vertical wavelengths and along-track horizontal wavenumbers are provided, as well as gravity wave temperature variances, potential energies and absolute momentum fluxes. This global data set captures the typical seasonal variations of these parameters, as well as their spatial variations. The GRACILE data set is suitable for scientific studies, and it can serve for comparison
- with other instruments (ground based, airborne, or other satellite instruments) and for comparison with gravity wave distributions, both resolved and parametrized, in GCMs and CCMs. The GRACILE data set is available as supplementary data at https://doi.org/10.1594/PANGAEA.879658.

#### 1 Introduction

Our work is focused mainly on the stratosphere and mesosphere, i.e. on the middle atmosphere in the approximate altitude 20 range from 20 to 90 km. In this altitude range typical scales of atmospheric gravity waves are from tens to a few thousand kilometers horizontally and from a few kilometers to several ten kilometers vertically (e.g., Preusse et al., 2008, and references

### Searth System Discussion Science Solutions Data

therein). Most gravity wave sources are located in the troposphere and lower stratosphere. The gravity waves seen higher up in the stratosphere and mesosphere have therefore mostly propagated upward from these sources. Some relevant sources are gravity waves excited by flow over topography (mountain waves) (e.g., McFarlane, 1987; Lott and Miller, 1997), gravity waves excited by convection (e.g., Fovell et al., 1992; Pfister et al., 1993; Piani et al., 2000; Song and Chun, 2005), and gravity waves generated by source processes related to strong wind jets, for example the subtropical jets or the polar jets (e.g.,

Plougonven and Zhang, 2014, and references therein).

Gravity waves propagate away from their sources. Thereby they redistribute momentum and energy in the atmosphere, and where they dissipate they can affect (accelerate or decelerate) the background flow by deposition of momentum and energy. Dissipation processes include radiative damping (e.g., Zhu, 1993), turbulent damping (e.g., Marks and Eckermann, 1995, and references therein), and wave saturation and breaking (e.g., Fritts, 1984; Fritts and Rastogi, 1985).

If a gravity wave propagates conservatively upward in a background atmosphere with constant background wind and temperature, its amplitude will grow exponentially due to the exponential decrease of atmospheric density with altitude. At some point, however, the amplitude reaches its saturation limit, and the wave will start to break. For an overview of the theory of wave saturation see, for example, Fritts (1984), or Fritts and Alexander (2003). Critical level filtering occurs when during wave

propagation the background wind is not constant and approaches the ground-relative phase speed  $c_{\varphi}$  of the wave. In this case, due to Doppler shifting, the intrinsic frequency  $\hat{\omega}$  and thus the vertical wavelength  $\lambda_z$  of the wave approach zero. Thereby also the saturation amplitude of the wave tends to zero, and the wave will dissipate completely. For a more detailed discussion see also Ern et al. (2015) and references therein.

One characteristic parameter of atmospheric gravity waves is  $E_0$ , the total gravity wave energy per unit mass:

$$20 \quad E_0 = E_{kin} + E_{pot} \tag{1}$$

with  $E_{kin}$  the kinetic, and  $E_{pot}$  the potential energy per unit mass. The kinetic energy is given by:

$$E_{kin} = \frac{1}{2} \left( \overline{u'^2} + \overline{v'^2} + \overline{w'^2} \right) \tag{2}$$

and the potential energy  $E_{pot}$  by:

$$E_{pot} = \frac{1}{2} \left(\frac{g}{N}\right)^2 \overline{\left(\frac{T'}{\overline{T}}\right)^2} \tag{3}$$

Here,  $\overline{T}$  is the atmospheric background temperature, g the gravitational acceleration of the Earth, and N the buoyancy frequency. Further, u', v', w', and T' are the perturbation components due to the gravity wave of the zonal, meridional and vertical wind, as well as the temperature, respectively. The overbar denotes averaging over one wave period or multiples of it.

Based on observed spectral characteristics, it is often assumed that the energy spectrum  $E(\mu, \hat{\omega}, \phi)$  of wind velocity or temperature perturbations due to gravity waves takes the form of a separable product of independent functions (e.g., Fritts and VanZandt, 1003; Fritts and Alexender, 2003);

$$E(\mu,\widehat{\omega},\varphi) = E_0 A(\mu) B(\widehat{\omega}) \Phi(\phi) \tag{4}$$

with  $\mu = m/m_*$  the ratio of gravity wave vertical wavenumber  $m = 2\pi/\lambda_z$  and the characteristic wavenumber  $m_*$  that separates the saturated from the unsaturated part of the vertical wavenumber spectrum. Often, the function  $A(\mu)$  is approximated as follows:

$$A(\mu) = \frac{A_0 \mu^s}{1 + \mu^{s+t}},$$
(5)

5 and  $B(\widehat{\omega})$  is often found to be proportional to  $\widehat{\omega}^{-p}$ :

$$B(\hat{\omega}) = B_0 \,\hat{\omega}^{-p} \tag{6}$$

 $A_0$  and  $B_0$  are normalization constants. The function  $\Phi(\phi)$  accounts for the anisotropy of the gravity wave distribution depending on the horizontal propagation direction  $\phi$ . The parameters *s*, *t*, and *p* are logarithmic spectral slopes. The spectral slope *s* describes the unsaturated part of the vertical wavenumber spectrum (at small *m*), and *t* the saturated part (at large *m*). While

- t = 3 is usually a very good approximation, s is not well constrained and often set to 1. The spectral slope p describes the shape of the intrinsic frequency spectrum B(\overline{\overline{\overline{\overline{\overline{\overline{\overline{\overline{\overline{\overline{\overline{\overline{\overline{\overline{\overline{\overline{\overline{\overline{\overline{\overline{\overline{\overline{\overline{\overline{\overline{\overline{\overline{\overline{\overline{\overline{\overline{\overline{\overline{\overline{\overline{\overline{\overline{\overline{\overline{\overline{\overline{\overline{\overline{\overline{\overline{\overline{\overline{\overline{\overline{\overline{\overline{\overline{\overline{\overline{\overline{\overline{\overline{\overline{\overline{\overline{\overline{\overline{\overline{\overline{\overline{\overline{\overline{\overline{\overline{\overline{\overline{\overline{\overline{\overline{\overline{\overline{\overline{\overline{\overline{\overline{\overline{\overline{\overline{\overline{\overline{\overline{\overline{\overline{\overline{\overline{\overline{\overline{\overline{\overline{\overline{\overline{\overline{\overline{\overline{\overline{\overline{\overline{\overline{\overline{\overline{\overline{\overline{\overline{\overline{\overline{\overline{\overline{\overline{\overline{\overline{\overline{\overline{\overline{\overline{\overline{\overline{\overline{\overline{\overline{\overline{\overline{\overline{\overline{\overline{\overline{\overline{\overline{\overline{\overline{\overline{\overline{\overline{\overline{\overline{\overline{\overline{\overline{\overline{\overline{\overline{\overline{\overline{\overline{\overline{\overline{\overline{\overline{\overline{\overline{\overline{\overline{\overline{\overline{\overline{\overline{\overline{\overline{\overline{\overline{\overline{\overline{\overline{\overline{\overline{\overline{\overline{\overline{\overline{\overline{\overline{\overline{\overline{\overline{\overline{\overlin}\overline{\overline{\overline{\overline{\overlin{\uverline{\overline{
- 15 or Ern et al. (2006) and references therein.

For a conservatively propagating gravity wave, however, the wave energy is not a conserved quantity. A parameter that is more relevant for the interaction of gravity waves with the background flow is the vertical flux of horizontal wave pseudomomentum. In the following, for simplification, we will call this parameter momentum flux. The momentum flux vector is given by:

20 
$$(F_{px}, F_{py}) = \overline{\varrho} \left(1 - \frac{f^2}{\widehat{\omega}^2}\right) \left(\overline{u'w'}, \overline{v'w'}\right)$$
 (7)

(e.g., Fritts and Alexander, 2003).  $F_{px}$  and  $F_{py}$  are the zonal and the meridional momentum flux components, respectively,  $\overline{\varrho}$  is the atmospheric background density, and f is the Coriolis parameter. This equation can be rewritten in terms of gravity wave wavenumbers and temperature amplitude (cf. Ern et al., 2004):

$$(F_{px}, F_{py}) = \frac{1}{2}\overline{\varrho} \left(\frac{g}{N}\right)^2 \frac{(k, l)}{m} \left(\frac{\widehat{T}}{\overline{T}}\right)^2$$
(8)

Here, T̂ is the temperature amplitude of the gravity wave, (k, l, m) = 2π (λ<sub>x</sub><sup>-1</sup>, λ<sub>y</sub><sup>-1</sup>, λ<sub>z</sub><sup>-1</sup>) is the wavenumber vector, consisting of zonal, meridional, and vertical component, respectively, with λ<sub>x</sub> and λ<sub>y</sub> the apparent horizontal wavelength in zonal (x) and meridional (y) direction, respectively, of a gravity wave with the "true" horizontal wavelength λ<sub>h</sub> in the direction of wave propagation. This equation was derived using the linear polarization relations for gravity waves (e.g., Fritts and Alexander, 2003; Ern et al., 2004). In Eq. (8) several terms were omitted for simplification. For the gravity waves seen by infrared (IR)
limb sounders, however, neglecting these terms introduces errors of only a few percent. For details see the discussion in the

supporting information of Ern et al. (2017). Equation (8) can be rewritten for absolute momentum fluxes  $F_{ph}$  by introducing the absolute horizontal wavenumber  $k_h = \sqrt{k^2 + l^2} = 2\pi/\lambda_h$ :

$$F_{ph} = \frac{1}{2}\overline{\varrho} \left(\frac{g}{N}\right)^2 \frac{k_h}{m} \left(\frac{\widehat{T}}{\overline{T}}\right)^2 \tag{9}$$

Similarly, the potential energy can be rewritten in terms of the gravity wave temperature amplitude with  $E_{pot,max}$  the maximum potential energy during one wave cycle:

$$E_{pot,max} = \frac{1}{2} \left(\frac{g}{N}\right)^2 \left(\frac{\widehat{T}}{\overline{T}}\right)^2 \tag{10}$$

and  $E_{pot}$  the potential energy of the wave averaged over one or more wave cycles:

$$E_{pot} = \frac{1}{4} \left(\frac{g}{N}\right)^2 \left(\frac{\widehat{T}}{\overline{T}}\right)^2 \tag{11}$$

which corresponds to Eq. (3).

The acceleration or deceleration (X, Y) of the background flow, in the following for simplification called gravity wave drag, is given by the vertical gradient of momentum flux:

$$(X,Y) = -\frac{1}{\overline{\varrho}} \frac{\partial(F_{px}, F_{py})}{\partial z}$$
(12)

with X and Y the drag in zonal and meridional direction, respectively, and z the vertical coordinate. For more details see the review paper by Fritts and Alexander (2003).

- Gravity wave drag plays an important role in the whole middle atmosphere. It significantly contributes to the wind reversals at the top of the mesospheric wind jets (e.g., Lindzen, 1981; Holton, 1982). Further, gravity wave dissipation drives the meridional circulation in the mesosphere, which leads to the cold summer mesopause, the coldest region in Earth's atmosphere, as well as to the relatively warm winter stratopause. Also in the stratosphere gravity wave drag plays an important role, for example for the driving of the quasi-biennial oscillation (QBO) and semiannual oscillation (SAO) of the zonal wind in the
- tropics (e.g., Lindzen and Holton, 1968; Dunkerton, 1997; Delisi and Dunkerton, 1988; Ern et al., 2014, 2015). In addition, gravity waves contribute to the Brewer Dobson circulation in the stratosphere, particularly to the summertime branch (e.g., Alexander and Rosenlof, 2003). A tutorial that addresses several effects of the interaction between gravity waves and the mean background flow is given in McLandress (1998).

Consequently, general circulation models (GCMs) and chemistry climate models (CCMs) need a realistic representation of gravity wave drag in order to produce realistic global circulation patterns in the middle atmosphere. The spatial resolution of these models, however, is usually too coarse to resolve more than a small fraction of the whole spectrum of gravity waves. Therefore most global models need gravity wave parametrization schemes (gravity wave drag schemes); see also McLandress (1998) or Kim et al. (2003) and references therein. At the time of writing, gravity wave parametrization schemes are still needed even for state-of-the-art high-resolution numerical weather prediction models (e.g., Orr et al., 2010), and also in the

30 foreseeable future gravity wave parametrization schemes will still be required.

Usually, gravity wave parametrization schemes launch gravity wave momentum flux from a source level, make assumptions about the propagation and dissipation of gravity waves, and thereby the effect (drag) that gravity waves exert on the background flow is simulated.

Traditionally, many global models employ at least two different gravity wave drag schemes: a nonorographic, and an orographic gravity wave drag scheme. Nonorographic gravity wave drag schemes usually do not represent specific gravity wave sources. Often, they assume a fixed source level and a homogeneous and isotropic launch distribution, i.e., they launch the same amount of momentum flux in different directions (for example, the four cardinal directions) at each model grid point. Some examples of such schemes are the schemes introduced by Lindzen (1981), Hines (1997), Alexander and Dunkerton (1999), Warner and McIntyre (2001), Scinocca (2003), or Yigit et al. (2008). Different from this, orographic gravity wave
parametrizations are dedicated to mountain waves that are excited by flow over topography, i.e. to a specific source process.

Some examples are McFarlane (1987), Lott and Miller (1997), or Scinocca and McFarlane (2000).

There are also attempts to address other specific sources by dedicated gravity wave parametrizations, for example, gravity waves excited by jets and fronts (Charron and Manzini, 2002; de la Cámara and Lott, 2015), or gravity waves excited by convective sources (e.g., Chun and Baik, 1998, 2002; Beres et al., 2004; Song and Chun, 2005; Bushell et al., 2015). These

15 schemes were successfully used in GCMs (e.g., Richter et al., 2010; Kim et al., 2013). Another recent development are socalled stochastic schemes (e.g., Eckermann, 2011; Lott et al., 2012; de la Cámara and Lott, 2015) which mimic the observed intermittent nature of gravity wave sources (e.g., Hertzog et al., 2008, 2012; Wright et al., 2013) in a simplified fashion.

Still, all these schemes are very simplified. They contain tunable parameters, make simplifying assumptions about the launch distributions, and most gravity wave drag schemes propagate gravity waves only in the vertical direction, while in a real atmo-

20 sphere gravity waves can also propagate horizontally (e.g., Marks and Eckermann, 1995; Sato et al., 2009, 2012; Preusse et al., 2009b; Ern et al., 2013; Kalisch et al., 2014; Hindley et al., 2015; Ribstein and Achatz, 2016). Therefore comparison with observed global distributions of gravity waves is important for improving and tuning gravity wave drag schemes. In particular, observed momentum fluxes allow for a direct comparison with gravity wave drag schemes.

There are already first attempts to improve gravity wave parametrizations by comparison with satellite observations. Some comparisons are based on gravity wave variances or amplitudes (e.g., Choi et al., 2009, 2012; Stephan and Alexander, 2015), while others are using momentum fluxes (e.g., Ern et al., 2006; Froehlich et al., 2007; Orr et al., 2010; Trinh et al., 2016; Kalisch et al., 2016).

Because these first comparisons have already led to promising results, the aim of our work is to provide a climatological data set GRACILE (= GRAvity wave Climatology based on Infrared Limb Emissions observed by satellite) of gravity wave

- temperature variances, squared temperature amplitudes, potential energies, horizontal wavenumbers, vertical wavelengths, and momentum fluxes based on three years (March 2005 until February 2008) of High Resolution Dynamics Limb Sounder (HIRDLS) observations, and on 13 years (February 2002 until January 2015) of Sounding of the Atmosphere using Broadband Emission Radiometry (SABER) observations. Both these instruments are infrared limb sounders operating on satellites in low Earth orbits. This measurement technique has the advantage that a comparably large range of the gravity wave spectrum is
- covered (see also Preusse et al., 2002, 2008; Alexander et al., 2010).

Of course, this climatological data set can also be used for comparison with distributions of gravity waves that are resolved in global models, in order to find out how realistic these distributions are. It has been shown that even for high resolution models gravity wave amplitudes may be underestimated, and distributions of resolved gravity waves may not be fully realistic (e.g., Schroeder et al., 2009; Preusse et al., 2014; Jewtoukoff et al., 2015). This means even distributions of resolved gravity waves need to be validated against observations. In addition, this climatological data set can be used for comparison with other observations, for example other satellite data, superpressure balloons, radiosondes, or ground-based instrumentation.

The manuscript is organized as follows: In Sect. 2 the HIRDLS and the SABER instruments are briefly introduced. Then, in Sect. 3, we describe how gravity wave temperature variances, potential energies and momentum fluxes are derived from temperature altitude profiles observed by HIRDLS and SABER. In addition, we address the observational limitations of the

10 instruments, and potential error sources are discussed. In Sect. 4, we describe how the data are gridded in preparation of the GRACILE climatological data set, and what data products are available. In particular, we present examples of global distributions, a measure of the natural variability, as well as time series of zonal averages. Finally, Sect. 5 gives a summary of the paper.

#### 2 The satellite instruments HIRDLS and SABER

- Our work is based mainly on data of the satellite instruments HIRDLS and SABER. Both instruments are infrared (IR) limb sounders operating on satellites in low Earth orbits. From atmospheric IR limb emissions of  $CO_2$  around 15  $\mu$ m temperaturepressure profiles of the atmosphere are derived. In addition, both instruments observe several trace species. In our study, we use HIRDLS version V006 (see also Gille et al., 2011) and SABER version v2.0 data. Detailed information about the HIRDLS instrument, temperature retrieval and vertical resolution is given, for example, in Gille et al. (2003), Gille et al. (2008),
- Barnett et al. (2008), or Wright et al. (2011). For SABER, details about the instrument are given, for example, in Mlynczak (1997), or Russell et al. (1999). The SABER temperature retrieval is described in Remsberg et al. (2004) and Remsberg et al. (2008).

HIRDLS observations are available from 22 January 2005 until 17 March 2008, while SABER observations started on 25 January 2002 and are still ongoing at the time of writing. However, in order to avoid biases in the GRACILE gravity
wave climatology, we use only full years of data. For HIRDLS, the GRACILE climatology covers March 2005 until February 2008, and for SABER February 2002 until January 2015. For an overview, Table 1 summarizes some characteristics of both instruments. Also given is the approximate temporal, latitudinal and altitude coverage of the observations, as well as the temporal and global coverage provided in our gravity wave climatology.

While HIRDLS continuously observes the latitude range of about 63°S–80°N, this is different for SABER: every ~60 days
for about 60 days SABER switches between a northward and a southward viewing mode with latitude coverages of 50°S–82°N and 82°S–50°N, respectively. This means that only the latitude range 50°S–50°N is observed continuously. For the range of years considered here (2002 until 2015), in February, June, and October the latitude coverage is always 50°S–82°N (northward view), and in April, August and December it is always 82°S–50°N (southward view). In the "odd" months (January, March,

May, July, September, and November) SABER switches between northward and southward view. Consequently, monthly averages of these months have a latitude coverage of  $82^{\circ}S-82^{\circ}N$ . However, latitudes poleward of  $50^{\circ}$  are only observed during part of the month, which may introduce biases in the gravity wave climatology poleward of  $50^{\circ}$  for those "odd" months.

Over the whole period of the SABER mission, the date when SABER switches between northward and southward view has 5 gradually shifted from the middle of the odd months to the beginning of the odd months. The first northward viewing phase of 2017 started even as early as 31 December 2016, i.e. not in January 2017.

#### **3** Satellite limb observations of gravity waves

Satellite instruments that observe Earth's atmosphere in limb geometry view toward the Earth horizon. A schematic of this viewing geometry is given in Fig. 1. Altitude profiles of the incoming limb radiances can be measured, for example, by
changing the elevation angle of the line of sight (LOS) of the instrument such that vertical scans through the atmosphere are performed. The point of the LOS that is closest to the Earth surface is the so called tangent point. In the case of optically thin emissions, most of the observed radiances have their origin in the vicinity of the tangent point, both in terms of altitude and in terms of horizontal position along the LOS. This is due to the exponential decrease of atmospheric density and, thus, the number of emitting molecules with altitude. Therefore, usually temperatures or trace gas mixing ratios that are derived from
observed altitude profiles can be attributed to the locations and altitudes (the "tangent altitudes") of the tangent points.

#### 3.1 Sensitivity function and observational filter

Limb sounding of optically thin atmospheric emissions is a measurement technique that is capable of observing small scale atmospheric fluctuations, such as gravity waves. This has first been reported by Fetzer and Gille (1994) and Eckermann and Preusse (1999). Later, Preusse et al. (2000) pointed out the importance of differences in the sensitivity of different measurement techniques for detecting gravity waves, and an analytic expression for the sensitivity function of limb sounders was derived (Preusse et al., 2002).

#### Sensitivity function:

The amplitude response  $S(k_{LOS}, m)$  of an altitude profile of observed limb radiances to an observed sine-shaped gravity wave due to effects of radiative transfer in the Earth atmosphere can be written as follows (Preusse et al., 2002):

$$S(k_{LOS},m) = \frac{1}{B} \frac{\partial B}{\partial T} \frac{\gamma^{1/2}}{(a^2 + \gamma^2)^{1/4}} \exp\left(\frac{-\gamma k_{LOS}^2}{4(\gamma^2 + a^2)}\right)$$
 (13)

Here, m is the vertical wavenumber, and k<sub>LOS</sub> = 2π/λ<sub>h,LOS</sub> the apparent horizontal wavenumber of the gravity wave in the direction parallel to the LOS of the instrument. In Fig. 2 an illustration is given showing that the apparent horizontal wavelength λ<sub>h,LOS</sub> parallel to the LOS, and the apparent horizontal wavelength λ<sub>h,AT</sub> parallel to the measurement track, can be quite different from the true horizontal wavelength λ<sub>h,true</sub> of an observed gravity wave. See also Preusse et al. (2009a) and
30 Trinh et al. (2015). The term <sup>1</sup>/<sub>B</sub> ∂B/∂T in Eq. (13) is the linear expansion in temperature T of the blackbody source function B.

Earth System Discussion Science science of Data

5

The further parameters in Eq. (13) are  $\gamma = 1/(2HR_{Earth})$ ,  $a = m/(2R_{Earth}) = \pi/(\lambda_z R_{Earth})$ , with  $R_{Earth}$  the Earth radius and H the pressure scale height. See also Preusse et al. (2002, 2008).

An ideal temperature retrieval (infinitesimal vertical field of view and infinitesimal retrieval step-width with at the same time infinite signal-to-noise ratio of the instrument) can compensate for effects of the vertical wavelength, but has to assume that an observed wave has infinite horizontal extent ( $k_{LOS} = 0$ ). The resulting temperature amplitude response of an ideal retrieval  $S_{T,ideal}(k_{LOS}, m)$  can be obtained by calculating the following ratio (Preusse et al., 2002, 2008):

$$S_{T,ideal}(k_{LOS},m) = S(k_{LOS},m)/S(k_{LOS}=0,m)$$
  
=  $\exp\left(\frac{-\gamma k_{LOS}^2}{4(\gamma^2+a^2)}\right)$  (14)

For a real retrieval, however, there will be a reduction of sensitivity at short gravity wave vertical wavelengths due to an
additional smoothing effect over an altitude interval Δz, caused by the vertical field of view of the instrument and the retrieval step-width. This smoothing effect can be accounted for by an additional contribution R(λz) (Preusse et al., 2002):

$$R(\lambda_z) = \frac{\lambda_z \sqrt{2}}{2\pi \Delta z} \sqrt{1 - \cos\left(\frac{2\pi \Delta z}{\lambda_z}\right)}$$
(15)

Usually, the vertical field of view of the instrument will dominate over the effect of the retrieval step, and can be set equal to  $\Delta z$ .

The sensitivity  $S_{T,real}(k_{LOS},m)$  of a "real" temperature retrieval to an observed gravity wave is then given by the product of  $R(\lambda_z)$  and  $S_{T,ideal}(k_{LOS},m)$  such that:

$$S_{T,real}(k_{LOS},m) = \frac{\lambda_z \sqrt{2}}{2\pi \Delta z} \sqrt{1 - \cos\left(\frac{2\pi \Delta z}{\lambda_z}\right)} \\ \times \exp\left(\frac{-\gamma k_{LOS}^2}{4(\gamma^2 + a^2)}\right)$$
(16)  
See clea Triph et al. (2015), their Eq. (1)

See also Trinh et al. (2015), their Eq. (1).

20 Relevant for our study is the sensitivity  $S_{A^2}(k_{LOS}, m)$  that is expected for gravity wave squared temperature amplitudes. This sensitivity also applies for gravity wave temperature variances, potential energies, or momentum fluxes. An analytic expression for  $S_{A^2}(k_{LOS}, m)$  is obtained by taking the square of  $S_{T,real}(k_{LOS}, m)$ :

$$S_{A^2}(k_{LOS},m) = S_{T,real}(k_{LOS},m)^2$$
(17)

In our study, we consider the satellite instruments HIRDLS and SABER that observe infrared limb emissions of atmospheric trace gases. For these instruments the analytic sensitivity function  $S_{A^2}(k_{LOS},m)$  is given as function of gravity wave horizontal and vertical wavelengths in Fig. 3a for HIRDLS, and in Fig. 3b for SABER by assuming vertical resolutions (vertical fields of view of the instruments) of 1 km for HIRDLS and 2 km for SABER, respectively. It should be pointed out that the horizontal wavelength relevant for the sensitivity function is the apparent horizontal wavelength of a gravity wave parallel to the line-of-sight direction of the satellite instrument (e.g., Preusse et al., 2009a). Compared to other global gravity wave ob-

30 servation techniques, limb sounding covers a quite large range of the gravity wave spectrum. See also Preusse et al. (2008) or Alexander et al. (2010).

# Science Science Science Science

5

We choose the parameters for the gravity wave analysis in a way that wave parameters for wavelengths shorter than 25 km are determined. In order to avoid that observed altitude profiles of temperature fluctuations are contaminated by gravity waves of longer vertical wavelengths, or with planetary waves, these altitude profiles are high-pass filtered in terms of vertical wavenumbers (see also Ern et al., 2011; Meyer et al., 2017). The resulting sensitivity is given in Fig. 3c for HIRDLS, and in Fig. 3d for SABER. The sensitivities shown in Figs. 3c and d are also provided in the GRACILE climatology together with the distributions of gravity wave parameters. It should however be pointed out that these sensitivities are just an approximation. The "true" sensitivity will be similar, but also depends on the details of the retrieval of temperatures from measured altitude profiles of atmospheric radiances. These retrieval details can lead to deviations from the idealized function  $S_{T,real}(k_{LOS}, m)$  (Preusse et al., 2002).

#### 10 **Observational filter:**

The analytic expression for the sensitivity  $S_{T,real}(k_{LOS},m)$  that combines the effects of radiative transfer, temperature retrieval and vertical field of view of the instrument (cf. Figs. 3a and 3b) already accounts for a major part of the overall observational filter of a limb sounding instrument. However, for the overall observational filter also other effects have to be taken into account. In particular, details of the wave extraction and wave analysis will have effect on the wave spectrum contained in

- 15 the temperature fluctuations that are attributed to gravity waves. For example, in our case an additional vertical filter was applied which modifies the sensitivity for gravity waves (cf. Figs. 3c and 3d). Further, if multiple altitude profiles are combined for the wave analysis, for example for deriving gravity wave momentum fluxes, also limitations of the spatial sampling of an instrument that lead to an undersampling of the horizontal structure of an observed gravity wave (aliasing) have to be considered (e.g., Ern et al., 2004; Trinh et al., 2015). Accounting for such effects is beyond the scope of our current study.
- 20 However, it has been shown by Trinh et al. (2015, 2016) that comparisons between observations and model data can be much improved if effects of the observational filter are taken into account by simulating the effect of the measurement and applying the simulated observational filter to the model data.

#### 3.2 Background removal

applied to extract the gravity wave signal.

The first step in any analysis of gravity waves from observations is the separation of the measured quantity into an atmospheric background and the fluctuations due to gravity waves. Particularly, temperature altitude profiles observed from satellite will contain contributions of both planetary waves with large horizontal scales and of gravity waves with much smaller horizontal scales. One of the major challenges of methods for removing the atmospheric background state from observed temperature altitude profiles is therefore to effectively separate the fluctuations due to planetary waves (which are usually much larger in amplitude) from those of gravity waves. Usually, this separation is done via a separation of scales, either vertically or horizontally. In the case of time series observed by ground based stations also temporal filtering of time series is frequently

Scale separation in vertical direction is usually performed by filtering observed altitude profiles vertically. One method is to use polynomial fits in the vertical direction as an estimate for the atmospheric background and subtract this background

# Science Scienc

from an altitude profile to obtain the fluctuations that are attributed to gravity waves. Another method is vertical filtering of single altitude profiles by introducing a low-pass filter for vertical wavelengths and attributing only fluctuations with vertical wavelengths shorter than about 10 km to gravity waves (e.g., Tsuda et al., 2000; de la Torre et al., 2006; Gavrilov, 2007). Scale separation in vertical direction works well in the wintertime polar lower stratosphere where vertical wavelengths of planetary waves are quite long, while those of gravity waves are usually much shorter. However, this approach has its shortcomings in

5 waves are quite long, while those of gravity waves are usually much shorter. However, this approach has its shortcomings in the tropics where planetary-scale equatorial wave modes and gravity waves generally have similar vertical wavelengths (e.g., Ern et al., 2008; Ern et al., 2014). Another general problem is that, by introducing a strong low-pass for vertical wavelengths, the remaining spectral range of gravity waves is considerably narrowed down.

Different from this, much of the vertical wavelength spectrum of gravity waves can be preserved if scale separation in 10 horizontal direction is utilized. Our approach of horizontal scale separation was introduced in Ern et al. (2011) and Ern et al. (2013). This approach aims at explicitly describing even day-to-day variations of the atmospheric background due to shortperiod traveling planetary waves, which is particularly important for investigating the gravity wave distribution in the tropics or in the mesosphere, but could also be relevant in the wintertime polar vortex because of its rapid temporal variations (e.g., Ern et al., 2016).

- The procedure utilized in our study for extracting small scale temperature fluctuations due to gravity waves from observed altitude profiles requires several steps. First, the zonal average background temperature is subtracted from each altitude profile of observed temperature. For estimating the contribution of planetary waves we calculate 2D spectra in longitude and time for overlapping time windows of 31 days length and a set of fixed latitudes and altitudes (Ern et al., 2011). Based on these spectra, the temperature perturbation due to planetary waves with zonal wavenumbers 1–6 and periods longer than about 1–2 days is
- calculated for the exact location and time of each observation in each altitude profile, and also subtracted. In this way, we even account for short-period planetary waves that can have periods as short as a few days, such as fast Kelvin waves in the tropics (e.g., Ern et al., 2008; Ern and Preusse, 2009), quasi two-day waves in the mesosphere (see also Ern et al., 2013), or short-period planetary waves in the wintertime polar regions (e.g., Ern et al., 2009; Ern et al., 2016). For removing tides, we utilize the fact that for satellites in slowly precessing low Earth orbits the ascending and descending nodes, respectively, are at about
- fixed local times. For HIRDLS, the local time does not change much during the mission, while for SABER the orbital plane slowly precesses (a full cycle is about 120 days). Consequently, tides will appear as stationary zonal wave patterns if data from ascending and descending nodes are taken separately. By removing these stationary wave patterns separately for ascending and descending nodes, tides can easily be removed from the observed temperature fluctuations (e.g., Preusse et al., 2001; Ern et al., 2013). In each altitude profile, we additionally remove the strongest oscillation with vertical wavelength of 40 km or longer in
- order to further suppress planetary waves, as well as long vertical wavelength gravity waves that are not covered by our method of determining gravity wave amplitudes. In addition, at altitudes above 60 km very short vertical wavelength oscillations in SABER altitude profiles are removed by a low-pass with a cutoff vertical wavelength of 5 km in order to remove oscillations that are presumably caused by minor retrieval artifacts in the mesopause region. On average, gravity wave vertical wavelengths are relatively long at these altitudes. Therefore, the effect of this additional filtering on the overall distribution of gravity waves
- should be small.

#### 3.3 Method for determining gravity wave amplitudes, phases, and vertical wavelengths

The resulting altitude profiles of temperature residuals are analyzed with a two-step method introduced by Preusse et al. (2002). First, the whole altitude profile is analyzed by the Maximum Entropy Method (MEM; Press et al. (1992)) for identifying all vertical wavelengths present in the profile. In the second step, in a sliding 10 km vertical window amplitudes and phases are

- fitted by a sinusoidal fit for all vertical wavelengths found by the MEM. For each altitude, the results are sorted according to the largest (second largest, and so on) amplitude. In the current paper, we further consider the strongest component only. Since the MEM is performed on the whole profile, we trust also wavelengths larger than the sliding window but not larger than approximately 25 km; therefore the filtering of removing all waves of 40 km and longer is applied. The resulting sensitivity functions combining both radiative transfer and retrieval effect as well as the vertical wavelength filtering are presented in
- Figs. 3c and 3d. The combination of MEM and sinusoidal fits, in short MEM/HA (HA for harmonic analysis) combines the advantages of addressing a relatively wide part of the vertical wavelength range and a fixed analysis window length. The latter is important, for instance, when investigating regions of wind shear where the vertical wavelength is refracted and strong gradients in wave amplitude are expected.

### 3.3.1 Latitude-altitude cross sections of gravity wave temperature variances, squared amplitudes and potential energies

The upper row of Fig. 4 shows latitude-altitude cross sections of zonal average gravity wave temperature variances for average January, April, July, and October determined from thirteen years of SABER data (February 2002 until January 2015). This time interval was used for all SABER latitude-altitude cross sections shown in our study. Temperature variances were multiplied by a factor of 2 to make them directly comparable to zonally averaged squared amplitudes that are also shown in Fig. 4. (Averaged

- over one wave period, the variance due to a perfect sine-wave is 0.5 times its amplitude squared.) The climatological cross sections shown in the first row of Fig. 4 represent the gravity wave temperature variances obtained directly after the removal of the atmospheric background state as described in Sect. 3.2, i.e. before the MEM/HA and the 10 km vertical windowing are applied. Overplotted contour lines represent the zonal average zonal wind of the Stratosphere-troposphere Processes And their Role in Climate (SPARC, A core project of the World Climate Research Programme) climatology for the respective month
- (see also Swinbank and Ortland, 2003; Randel et al., 2002, 2004).

The dominant climatological features are an overall increase of gravity wave temperature variances with altitude, which is expected due to the decrease of atmospheric density with altitude. Further, temperature variances are particularly enhanced in the polar region during wintertime, which is caused by strong activity of orographic and polar-jet related gravity wave sources. In addition, the strong background wind offers favorable propagation conditions (increased saturation amplitudes) for gravity

waves propagating opposite to the background winds. Another enhancement of temperature variances is seen in the summertime subtropics, which is mainly caused by gravity waves excited by convective sources and favorable propagation conditions in the subtropical jets. These features are qualitatively in good agreement with several previous studies (e.g., Fetzer and Gille, 1994; Wu and Waters, 1996; Jiang et al., 2004; Alexander et al., 2008; Ern et al., 2011).

5

The second row in Fig. 4 shows the corresponding squared amplitudes for the strongest wave component obtained by applying the MEM/HA and 10 km vertical windowing. The distributions are almost the same as for gravity wave temperature variances times a factor of two, only absolute values are somewhat reduced for squared amplitudes. This reduction is caused by the fact that we consider only the strongest wave component at each altitude and neglect smaller amplitude waves that will also exist (e.g., Wright and Gille, 2013). However, the contribution of those higher-order small-amplitude waves to both squared amplitudes and momentum fluxes is usually small, and their distribution is easily biased by instrument noise and other instrument effects.

The third row in Fig. 4 shows gravity wave squared amplitudes of those pairs of altitude profiles that are considered suitable for the determination of momentum fluxes (i.e. those pairs of altitude profiles with matching gravity wave vertical wavelength

- 10 and at the same time short enough horizontal sampling distance, see also Sect. 3.4 below). As can be seen from Fig. 4, squared amplitudes considering all altitude profiles are almost exactly the same as the squared amplitudes of the pairs of altitude profiles used for momentum flux determination. This indicates that these "suitable" pairs should be still representative for the global distribution of gravity waves.
- Figure 5 shows the same as Fig. 4, but for the HIRDLS instrument and the corresponding three-year time period (March
  2005 until February 2008). This time interval was used for all HIRDLS latitude-altitude cross sections shown in our study.
  SABER and HIRDLS distributions are very similar. Even the absolute values are in good agreement. Minor differences may arise from differences in the viewing geometries (different line-of-sight directions and different vertical field of view), or from (minor) differences of the "real" instrument sensitivity functions caused by differences in the temperature retrieval.
- Once, gravity wave temperature variances or squared amplitudes are available, the determination of potential energies is straightforward by applying Eq. (3) for gravity wave temperature variances, or Eq. (11) for squared amplitudes. Similar as Figs. 4a–d, Figs. 6a–d show zonal average cross sections of gravity wave potential energies calculated from SABER temperature variances for the average months of January, April, July, and October. Figs. 6e–h show the same, but for the HIRDLS instrument. As expected, the basic features of the distributions displayed in Fig. 6 are the same as in Figs. 4 and 5. Also available as part of the GRACILE gravity wave climatology are zonal average distributions for the other average calendar months.
- 25 All gravity wave potential energy values given in the climatology are calculated directly from temperature variances using Eq. (3). This means that no 10 km vertical window is applied, and values represent averages over a full wave cycle.

#### **3.3.2** Error considerations

Gravity waves appear as temperature fluctuations in observed altitude profiles. Accordingly, systematic errors of the temperature retrieval are removed by the separation into gravity wave fluctuation and background. This holds both for constant

30 offsets as well as for offsets slowly varying with geolocation (e.g. offsets dependent on altitude or latitude). Different from this, measurement noise leads to random temperature fluctuations that will affect the estimation of gravity wave temperature variances and squared amplitudes. Estimates of the temperature precision are given, for example, by Gille et al. (2011) for HIRDLS and are also provided for each HIRDLS altitude profile together with the temperature data. Therefore, it is possible to compare HIRDLS random errors directly with the estimated gravity wave temperature variances. For SABER,

25

the temperature precision was estimated by Remsberg et al. (2008), and values are also given on the SABER website at http://saber.gats-inc.com/temp\_errors.php. In Table 2 we have summarized these SABER precision estimates. In Table 2, temperature standard deviations, as well as variances (standard deviations squared), are given for "normal" midlatitude conditions, as well as for conditions of a cold summer mesopause.

- In order to find out whether random errors may affect the determination of gravity wave temperature variances or amplitudes, Fig. 7 shows zonal average cross sections of the ratio of temperature precision squared (random error variances) to gravity wave temperature variances after background removal for the average month of January (left column), April (second column), July (third column), and October (right column). The upper row is for SABER, and the lower row for HIRDLS. Overplotted contour lines in Fig. 7 represent temperatures for the respective month taken from the SPARC climatology (Randel et al., 2002, 2004).
- Cross sections for each average calendar month are provided as part of the GRACILE gravity wave climatology. For the climatology, SABER random error variances for cold mesopause conditions during austral summer are adopted poleward of 50° S for the months of November until February with a smooth transition to wintertime random error variances north of 40° S. Similarly, during boreal summer, SABER random error variances for cold mesopause conditions are adopted poleward of 50° N for the months of May until August with a smooth transition to wintertime random error variances south of 40° N.
  For all other conditions wintertime random error variances are assumed for SABER.
- For HIRDLS the precision (random error) predicted by the retrieval algorithm is provided together with each retrieved temperature profile. As stated in Gille et al. (2011), these theoretical values should be an upper estimate because the temperature precision estimated directly from retrieved HIRDLS temperature profiles in regions of low atmospheric variability is better than the theoretical estimate by about a factor of two (Gille et al., 2011, their Fig. 5.1.3). Therefore, for the HIRDLS values shown in
- Fig. 7, as well as for the values provided together with the gravity wave climatology, we used values of the predicted HIRDLS precision (standard deviation) divided by two.

Error estimates are, of course, uncertain to some degree and we here compare zonal mean values of gravity wave temperature variances, which are averages over strong and weak gravity wave events. Therefore even in regions where on average the fraction of noise is very small, noise may still influence the results via the weak events to some degree. On the other hand, we are using the strongest component only, which suppresses noise in the presence of a real wave.

As can be seen from Fig. 7, gravity wave temperature variances usually are well above the noise level. There are only two exceptions: the summertime high latitudes in the lower and middle stratosphere, and the cold summer mesopause region. In particular, in the summer mesopause region considerable biases should be expected. In this region, the temperature precision is about 7 K, which corresponds to about 50% of the estimated variances in Figs. 4a and 4c. Therefore, gravity wave temperature

30 variances and squared amplitudes, potential energies, and momentum fluxes will be high-biased. This has already been pointed out by Ern et al. (2011): in this region their wave analysis showed phase differences between pairs of altitude profiles that were indicative of an enhanced noise level.