# Peer review of "GRACILE: A comprehensive climatology of atmospheric gravity wave parameters based on satellite limb soundings"

_Earth System Science Data, 2017_

## Referee Comment (RC1) · Anonymous Referee #1 · 28 Jan 2018

This paper gives a very comprehensive presentation on the background, data sources, methodology, and error estimates for GRACILE (GRAvity wave climatology based on Infrared Limb Emissions observed by satellite) climatology data set on gravity waves in the stratosphere and mesosphere. It gives monthly values for gravity wave temperature radiances and squared amplitudes and the gravity wave potential energies, vertical wavelengths, horizontal wavenumbers, and absolute momentum fluxes derived from the limb scanning measurements from the HRDLS and SABER satellite instruments. All of these GRACILE monthly averages are available through the open access world data center PANGEA.

The background discussion in this paper is quite complete except for a few items. One is on lines 5-6 on page 6. Here, the authors point out the usefulness of this data set for comparison with other measurements, but they don't caution that every observation method has its own coverage in gravity wave wavenumber and frequency space, and that needs to be taken into account. It is also somewhat odd that on lines 19-22, they say that discussion of the vertical filter used for these data is "beyond the scope of the current study." I think a short paragraph summarizing those effects (with references) would be welcomed by the reader. Another point that could benefit from further discussion is that on line 6 of page 11. In Geller et al. (2013), on page 6387, there is a discussion on how data retention affects derived gravity wave momentum fluxes. That discussion contrasts the two methods used for deriving momentum fluxes from HRDLS used in that paper. The authors should point out how their data selection relates to the discussion in Geller et al. (2013). I believe that the statement on lines 12-13 on page 12 also relates strongly to this discussion in Geller et al. (2013). The discussion on lines 10-15, on page 13, leaves the reader wondering why this was done. Please explain the reasoning for this.

The paper then goes on to show latitude-altitude cross-sections of the various parameters in GRACILE. There are a few things about those figures, and the accompanying discussion, where I think further comment is needed. In figure 9, the HRDLS vertical wavelengths look longer. I find the discussion on lines 29 on page 15 to line 2 on page 16 to be confusing on this issue. Would the authors please work to make their points clearer on this issue. I don't understand why results for vertical wavelengths are shown, but results for horizontal wavenumbers are shown. Unless there is a good reason for this, I urge their results be shown for wavelengths in both cases unless the authors have a good reason for showing wavelengths in one case and wavenumbers in the other. If there is such a good reason , the authors should give their explanation. The reduced values of kh at low latitudes and at higher altitudes has been previously noted by Wang et al. (2005, J. Atmos. Sci.), albeit from radiosonde data in the troposphere and stratosphere. In general, it would be good if the authors noted where their results

are consistent, or inconsistent, with works using different techniques and thus sensitive to different portions of the gravity wave spectrum. The statement on lines 19-20 of page 17 is a good one, but it should be reinforced by saying that, for that reason, the vertical derivative of the gravity wave momentum fluxes from GRACILE are likely not indicative of quantitative mean flow accelerations due to this. In the discussion of figure 12, no mention is made of the HIRDLS/SABER differences. The HIRDLS momentum fluxes look larger to me than those from SABER where they overlap. The short paragraph on lines 6-11 on page 18 might say more about the work of Trinh et al. (2016) who wrote a paper on this subject. The HIRDLS/SABER differences in figure 10 are quite large. The authors indicate the results are unreliable in some regions. Is this their explanation? The statement on lines 30-33 on page 21 is rather unsatisfactory. Why do the authors think the offsets are "minor?" Again, on lines 20-21 on page 22, the offsets are relatively small, but they look systematic, not indicative of random error. In general, it is my impression that the authors tend to downplay HIRDLS/SABER differences too much. It would be better if they indicated what the readers should quantitatively trust and what should be more qualitatively trusted.

I find it odd that, while the paper by Meyer et al. (2017) is mentioned in line 4 on page 9, nowhere do I remember seeing a statement that the satellite limb scanning gravity waves will not be seeing waves that comprise much, if not most, of the gravity wave momentum fluxes in many regions. I think this needs to be said. This does not detract from the value of the GRACILE dataset, but this should be explicitly pointed out.

In general, this paper is very well written, but nonetheless, I do have a few detailed comments which follow.

1. Page 4, line 18: …., in the stratosphere, … 2. Page 12, line 23: "As expected" is an understatement. "As must be the case" is more appropriate. 3. Page 16, line 9: It's not that the limitation is "more relaxed." Rather, the Coriolis parameter is smaller so there is more space between the two limitations of the Coriolis parameter and the Brunt Vaisala frequency. 4. Page 22, line 14: What is GLIGOSS? 5. The statement

of a likely solar cycle in gravity wave parameters in GRACILE is very weak, given the length of measurement. Perhaps, just point out what Li et al. (2016) have said.

---

## Referee Comment (RC2) · Anonymous Referee #2 · 27 Feb 2018

Review comments for "GRACILE: a comprehensive climatology of atmospheric gravity wave parameters based on satellite limb soundings" by Ern et al.

This paper thoroughly describes an atmospheric gravity wave (GW) dataset that consists with two satellite instrument measurements: SABER and HIRDLS. This is a Level-3 dataset that has already been gridded and monthly-averaged. Both instruments have similar viewing geometry so their observed GW parameters are highly consistent with each other. This dataset provides a valuable quantitative estimations of stratospheric GW parameters for users to compare with their measurements or model outputs.

This dataset was made possible from the authors' group's many years of scientific

exploration of studying GW characteristics using SABER and HIRDLS, so the quality is very trustworthy. As an Algorithm Theoretical Basis Document (ATBD) type of manuscript, the descriptions of the methodology, advantage and disadvantage of each derived variables are very clear. I suggest publication after some minor revision (typos, image edit, etc.)

(1) One major concern is about how to grid the data. Gridding highly intermittent features such as the atmospheric gravity wave requires the denominator to be the number of pass-by observations during a month. For example, monthly mean GWMF in the gridbox [5W,5S, 5E,5N] equals total of all retrieved GWMF values within this gridbox divided by all pass-by samples fall into this gridbox during the month. So when you use paired obs. for calculation and if they fall into two grid boxes, which one do you assign it? I'm not saying you are wrong, it's just not very clear to me in the description of gridding methodology.

(2) Another concern is that, from Fig. 15 (c) I can see two stripes of enhancement of momentum flux samples around 50S and 50N. I suspect that's partly caused by the fact that GWs there are strong during wintertime, so you have greater chance to find pairs to complete your GWMF calculation. But I don't see such hints in HIRDLS map in Fig. 15 (f). Why that's the case? If my interpretation was wrong, then what causes the enhancements at 50S and 50N in Fig. 15 (c)?

Minor points: Page 3, Line 13: please considering adding one more recent reference that has validated the theoretical value proposed by van Zandt, 1985 from an observation or multiple different observations.

Page 5, Line 34: how do you read the new "observational filter" in de le Torre et al. (2018)? In their paper, they sort of suggests that SABER's observational window is very narrow.

de la Torre, A., Alexander, P., Schmidt, T., Llamedo, P., and Hierro, R.: On the distortions in calculated GW parameters during slanted atmospheric soundings, Atmos.
Meas. Tech. Discuss., https://doi.org/10.5194/amt-2017-192, in review, 2017.

Page 6, Line 4: please consider also including Gong et al., 2015 in the reference Gong, J., J. Yue, and D. L. Wu (2015), Global survey of concentric gravity waves in AIRS images and ECMWF analysis. J. Geophys. Res. Atmos., 120, 2210–2228. doi: 10.1002/2014JD022527.

Page 13, Line 21: I didn't read Gill et al., 2011, so can't comment further, but arbitrarily divide every number by two seems dangerous to me. Can you provide a rough picture like how many observed samples are $\frac{1}{2}$ of theoretical value and what is the standard deviation?

Page 15, Line 29: you may want to add "except inside the jet streams".

Page 20, paragraph 1: so GRACILE only starts from z=30 km, correct? If that's the case, I strongly suggest you change Fig. 15 (I'll mention my suggestion when comes to that point).

Page 20, Line 13: that's where my major concern #2 comes from. If you can draw a lat-lon map you can discover whether the enhancement of sample size is source-related. Also, it would be helpful to add a bit discussion here if that's the case. Since stronger source gives you larger chance to "see" them because it's easier to find pairs.

Page 20, Line 31: do you provide a quality flag for each value in each grid box? So user can easily make their own plots according to their needs.

Page 22, Line 14: GLIGLOSS -> GRACILE?

Fig. 10: please consider use horizontal wavelength as the colorbar so the unit and physical meaning would be consistent with Fig. 9 (or the other way around).

Fig. 15: Since the sample size is not height dependent, I strongly suggest you make only line plot so you only need two plots (one for SABER with three lines as a function of latitude, and one for HIRDLS) to explain the sample size matter. The only exception is

at ∼ 20 km for HIRDLS tropics. But since you don't provide data below 30 km because of potential cloud contamination, no need to show and discuss about that anyway.

Fig. 18: Instead of using dashed lines to show the natural variability, I think you can use semi-transparent grey/color areas to enclose the natural variability.

―――――――――――――

---

## Author Comment (AC1) · 26 Mar 2018

Dear Referee # 1,

We greatly appreciate your detailed and constructive comments. Based on these comments the manuscript was significantly improved. In particular, we were made aware that in some places more background information was needed.

Here first we will give short responses to the main concerns. A more detailed response will be given in the next stage of the review in the detailed point-by-point reply and the revised manuscript.

[Figure]

In the following, reviewer comments are given in black, our responses are given in blue.

Again, thank you very much for your detailed review!

**Minor Comments**

(1) lines 5-6 on page 6.
Here, the authors [...] don't caution that every observation method has its own coverage in gravity wave wavenumber and frequency space, and that needs to be taken into account.

We have added some text stating that for a meaningful comparison of different instruments their respective observational filter has to be taken into account.

(2) [page 9] on lines 19-22, they say that discussion of the vertical filter used for these data is "beyond the scope of the current study." I think a short paragraph summarizing those effects (with references) would be welcomed by the reader.

The statement "beyond the scope of the current study." did not refer to the vertical resolution of the instruments, but to the possibility of horizontal wave structures being undersampled by the horizontal sampling step of the instrument (aliasing effect). An empirical correction for those aliasing effects was suggested by Ern et al. (2004).

This correction, however, was based on assumptions on the shape of the wave spectrum in a given region — an assumption that we wanted to avoid, and our statement "Accounting for such effects is beyond the scope of our current study." referred to the non-application of this correction in our current work. We also did not correct for effects of the sensitivity function.

Of course, the reviewer is correct that some more background information is needed

here. Therefore we added a detailed discussion in the revised manuscript, and the end of Sect. 3.1 was partly rewritten.

(3) line 6 of page 11. In Geller et al. (2013), on page 6387, there is a discussion on how data retention affects derived gravity wave momentum fluxes. That discussion contrasts the two methods used for deriving momentum fluxes from HRDLS used in that paper. The authors should point out how their data selection relates to the discussion in Geller et al. (2013). I believe that the statement on lines 12-13 on page 12 also relates strongly to this discussion in Geller et al. (2013).

(3a) About page 11, line 6:
In this line we state that in an altitude profile, at each given altitude, we consider only the gravity wave with the strongest amplitude. This assumption is common to both methods of deriving gravity wave momentum flux ("HIRDLS1" and "HIRDLS2") that are presented in Geller et al. (2013). (For the "HIRDLS1" method see Alexander et al. (2008), the text related to their Eq. (5).) As has been discussed in more detail later in our Sect. 3.3.1, this assumption is justified because smaller-amplitude waves do not contribute much to the overall temperature variance. This can be seen by comparing rows 1 and rows 2 in Fig. 4 (for SABER) or rows 1 and rows 2 in Fig. 5 (for HIRDLS). In both these rows all altitude profiles are considered (i.e., no "data retention"). We find that squared temperature amplitudes are only somewhat lower than two times the temperature variance, which means that higher-order waves do not contribute much.

In the revised manuscript we now state that this assumption has been made also by other authors who derive gravity wave momentum fluxes using another approach. Further, we state that higher-order waves do not contribute much and refer to the later Sect. 3.3.1.

(3b) About page 12, lines 12–13:

Indeed, the case of non-matching gravity waves in a pair of considered altitude profiles is an issue that is treated differently in the two methods of deriving gravity wave momentum fluxes from satellite data that are presented in Geller et al. (2013).

The "HIRDLS1" method attributes small amplitudes and little momentum fluxes to non-matching pairs, even in cases when the amplitudes in both profiles of a pair are large, while in our method ("HIRDLS2") non-matching pairs are not considered for the momentum flux calculation.

Regarding average momentum fluxes calculated in a certain region, the first method will result in much lower average values than the second method. The second method inherently assumes that the matching pairs are representative for the average momentum flux in this region. Figs. 4 and 5 provide evidence supporting this assumption: the whole number of single profiles and the reduced number of matching pairs of altitude profiles, and thus also the not-selected pairs, have almost the same distribution and magnitude of gravity wave squared amplitudes.

A detailed discussion was added after former l.7 on p.12.

(4) The discussion on lines 10-15, on page 13, leaves the reader wondering why this was done. Please explain the reasoning for this.

As stated in the manuscript, our purpose is to provide cross sections of temperature random error variances for all calendar months. SABER random error variances are however given only for the two cases of "normal" conditions and "cold summer mesopause" conditions. We therefore used "cold summer mesopause" estimates for those months and latitudes where the SPARC climatology indicates cold summer mesopause temperatures (usually poleward of 50deg in the summer hemisphere during these months) and "normal" estimates elsewhere. In order to avoid jumps, a smooth transition between 40deg and 50deg latitude was introduced, where applicable. This is now written more clearly in the revised manuscript.

(5) In figure 9, the HRDLS vertical wavelengths look longer. I find the discussion on lines 29 on page 15 to line 2 on page 16 to be confusing on this issue. Would the authors please work to make their points clearer on this issue.

This confusion is probably caused by mixing two different effects in the same paragraph. The paragraph has been split and the content moved to where it fits better. Partly, the text has been rewritten. For changes see the revised manuscript.

(6) I don't understand why results for vertical wavelengths are shown, but results for horizontal wavenumbers are shown. Unless there is a good reason for this, I urge their results be shown for wavelengths in both cases unless the authors have a good reason for showing wavelengths in one case and wavenumbers in the other. If there is such a good reason , the authors should give their explanation.

The reason for showing horizontal wavenumbers instead of horizontal wavelengths is that the range of vertical wavelengths is limited to below ~25km due to the used vertical analysis interval and sensitivity function. Horizontal wavelengths, however, are not limited and single values can attain very large values of a few thousand km. Showing average horizontal wavelengths would therefore overemphasize those values that do not contribute much to average momentum fluxes and that therefore are not representative for the average distribution of gravity wave momentum fluxes.

This is now stated after former line 3 on page 16. For convenience, additional horizontal wavelength scales have been added in Figs. 10, 16, and 17.

(7) reduced values of kh at low latitudes and at higher altitudes has been previously noted by Wang et al. (2005, J. Atmos. Sci.), albeit from radiosonde data in the troposphere and stratosphere. In general, it would be good if the authors noted where their results are consistent, or inconsistent, with works using different techniques and thus sensitive to different portions of the gravity wave spectrum.

As recommended, the reference Wang et al. (2005, J. Atmos. Sci.) has been added.

(8) The statement on lines 19-20 of page 17 is a good one, but it should be reinforced by saying that, for that reason, the vertical derivative of the gravity wave momentum fluxes from GRACILE are likely not indicative of quantitative mean flow accelerations due to this.

For two reasons, we keep the statement as is:

First, the reason for the discrepancy of momentum flux vertical gradients between models and observations as discussed in Geller et al. (2013) is still not clear and could be an effect of the measurements, the models, or of both.

Second, the discussion in Geller et al. (2013) addresses an overall "weak" vertical gradient in momentum fluxes based on globally averaged data. Accordingly, gravity wave drag resulting from those gradients would be also weak. This situation is much different from "localized" phenomena, such as the strong gradients at the top of strong wind jets. In such situations, vertical gradients in observed momentum fluxes will still provide valuable information on the forcing of the background flow, as has been shown, for example, for the summertime mesospheric jets, the QBO, the SAO, and the wintertime polar vortex (Ern et al., 2013, 2014, 2015, 2016). For these situations, observations mostly show good agreement with model results.

(9) In the discussion of figure 12, no mention is made of the HIRDLS/SABER differences. The HIRDLS momentum fluxes look larger to me than those from SABER where they overlap.

At the end of Sect. 3.4.3, we have added the information that HIRDLS values of gravity wave momentum flux are somewhat higher in the polar vortices. One possible reason is that in these regions average horizontal wavelengths are relatively short (cf. Fig. 10).

Accordingly, the better HIRDLS along-track sampling will lead to reduced aliasing effects compared to SABER and result in higher momentum fluxes.

(10) The short paragraph on lines 6-11 on page 18 might say more about the work of Trinh et al. (2016) who wrote a paper on this subject.

As recommended, at the end of Sect. 3 we have now summarized the different effects of the observational filter that are mentioned in Trinh et al. (2015, 2016) and that have to be taken into account for measurement/model comparisons. For details see the revised manuscript.

(11) The HIRDLS/SABER differences in figure 10 are quite large. The authors indicate the results are unreliable in some regions. Is this their explanation?

Of course, there are some regions where horizontal wavenumbers are not very reliable. These regions are discussed in detail on former page 16, lines 19–32.

The general difference between HIRDLS and SABER, however, is another effect. SABER has a coarser sampling step and will therefore horizontally undersample a larger number of short horizontal wavelength waves than HIRDLS, resulting in lower horizontal wavenumbers on average.

This issue was addressed on former page 16, lines 12–14, however, it should indeed be pointed out more clearly that these lines address the main difference between HIRDLS and SABER. This paragraph has been reworded accordingly. Further, we refer to the discussion about aliasing effects that was newly introduced at the end of Sect. 3.1. See also our reply to Reviewer # 1, main comment (2).

(12) The statement on lines 30-33 on page 21 is rather unsatisfactory. Why do the authors think the offsets are "minor?"

Considering an overall error of momentum fluxes of a factor of two or more, as stated in Sect. 3.4.4, the differences between SABER and HIRDLS momentum fluxes can be considered small. The statement has been reworded accordingly.

(13) Again, on lines 20-21 on page 22, the offsets are relatively small, but they look systematic, not indicative of random error. In general, it is my impression that the authors tend to downplay HIRDLS/SABER differences too much. It would be better if they indicated what the readers should quantitatively trust and what should be more qualitatively trusted.

We are sorry, we did not want to downplay HIRDLS/SABER differences. Our statement on page 22, lines 20/21 was intended to remind the reader that the small differences between SABER and HIRDLS should not be taken as a measure of the overall uncertainty of the momentum flux values presented, which is still a factor of two or more, as indicated in lines lines 20/21. This is now stated more clearly in the revised manuscript.

(14) I find it odd that, while the paper by Meyer et al. (2017) is mentioned in line 4 on page 9, nowhere do I remember seeing a statement that the satellite limb scanning gravity waves will not be seeing waves that comprise much, if not most, of the gravity wave momentum fluxes in many regions. I think this needs to be said. This does not detract from the value of the GRACILE dataset, but this should be explicitly pointed out.

Yes, indeed, this is an important point that should be more emphasized. On former page 9 we have therefore added a few sentences to emphasize this point.

**Technical Comments**

(1) Page 4, line 18: ..., in the stratosphere, ...

We omitted "also" because it is not needed here, and started the new sentence with:

"In the stratosphere,..."

(2) Page 12, line 23: "As expected" is an understatement. "As must be the case" is more appropriate.

Changed as requested.

(3) Page 16, line 9: It's not that the limitation is "more relaxed." Rather, the Coriolis parameter is smaller so there is more space between the two limitations of the Coriolis parameter and the Brunt Vaisala frequency.

As suggested, the sentence has been reworded as follows:

"This effect is caused by the fact that in the tropics the Coriolis parameter is smaller, i.e., there is more space between the two limitations of the Coriolis parameter and the buoyancy frequency and longer horizontal wavelength gravity waves can exist."

(4) Page 22, line 14: What is GLIGOSS?

Thank you very much! Acronym has been corrected!

(5) The statement of a likely solar cycle in gravity wave parameters in GRACILE is very weak, given the length of measurement. Perhaps, just point out what Li et al. (2016) have said.

Also Li et al. (2016) discussed a correlation of gravity wave activity with the solar cycle. Of course, similar as SABER, the radiosondes used by Li et al. (2016) cover only somewhat more than one 11-year solar cycle. Therefore our statement has been downtoned, and we mention that the data sets are relatively short:

"In addition, there is a weak quasi-decadal variation (see also Ern et al., 2011). Similar quasi-decadal variations are also found in gravity wave energy densities observed by radiosondes (Li et al., 2016). These variations might be correlated with the 11-year solar cycle, however, much longer data sets would be needed for an in-depth investigation of this effect."

---

## Author Comment (AC2) · 26 Mar 2018

Dear Referee # 2,

Thank you very much for carefully reading our paper! Based on your comments the manuscript could be improved by, for example, a better description of several aspects of the data processing. Further, it was clarified that vertical wavelength biases as discussed by de la Torre et al. (2018) do not play an important role for the data presented.

Here first we will give short responses to the your concerns. A more detailed response will be given in the next stage of the review in the detailed point-by-point reply and the

revised manuscript.

In the following, reviewer comments are given in black, our responses are given in blue.

Again, thank you very much for your careful review!

**Main Concerns**

(1) One major concern is about how to grid the data. Gridding highly intermittent features such as the atmospheric gravity wave requires the denominator to be the number of pass-by observations during a month. For example, monthly mean GWMF in the gridbox [5W,5S, 5E,5N] equals total of all retrieved GWMF values within this gridbox divided by all pass-by samples fall into this gridbox during the month. So when you use paired obs. for calculation and if they fall into two grid boxes, which one do you assign it? I'm not saying you are wrong, it's just not very clear to me in the description of gridding methodology.

Indeed, some care has to be taken when calculating averages. For clarification, we have added the following text in Sect. 4.2 where the gridding is described.

"A monthly mean value assigned to a gridbox equals the total of all values within this gridbox divided by the number of all data points within the gridbox. Each "paired observation" is treated as a new data point, and the center coordinates between the two single observations that contribute to this paired observation are taken as the new coordinates for the pair, i.e., we assign new coordinates in latitude, longitude and time to the pair. In this way, ambiguities are avoided at the cost of creating a new set of coordinates."

(2) Another concern is that, from Fig. 15 (c) I can see two stripes of enhancement of momentum flux samples around 50S and 50N. I suspect that's partly caused by the fact that GWs there are strong during wintertime, so you have greater chance to find pairs to complete your GWMF calculation. But I don't see such hints in HIRDLS map in Fig. 15 (f). Why that's the case? If my interpretation was wrong, then what causes the enhancements at 50S and 50N in Fig. 15 (c)?

The enhancements of the number of points available are just an effect of the satellite sampling geometry, and not an effect of the pair-selection for determining gravity wave momentum fluxes. This is evident already from the left and middle columns in Fig. 15 that show an enhanced number of samples at the same latitudes as in the panels of the right column. The left and middle columns are for single altitude profiles, i.e., selection of pairs was not applied. Further, as has been shown in Fig. 8, the rate at which pairs are selected is mostly between 55 and 65% and does not show much latitudinal variation.

The basic reason for enhancements in the measurement density is that during one orbit a satellite in low Earth orbit spends more time at the turning points (high latitudes) of the orbit than at low latitudes.

A more detailed discussion is given in the detailed point-by-point reply. Further, we have added some explanation in the revised manuscript in the middle of former page 20.
**Minor Comments**

(1) Page 3, Line 13: please considering adding one more recent reference that has validated the theoretical value proposed by van Zandt, 1985 from an observation or multiple different observations.

We have added the reference Placke et al. (2013) for the mesosphere (a combination of radar and lidar observations), and the reference Hertzog et al. (2002) for the lower stratosphere (superpressure balloon observations), as well as Tsuda et al. (2000) and Nastrom et al. (2000) using a combination of radar and GPS-RO.

Hertzog, A., Vial, F., Mechoso, C. R., Basdevant, C., and Coquerez, Ph.: Quasi-Lagrangian measurements in the lower stratosphere reveal an energy peak associated with near-inertial waves, Geophys. Res. Lett., 29, 1229, doi:10.1029/2001GL014083, 2002.

Nastrom, G. D., Hansen, A. R., Tsuda, T., Nishida, M., and Ware, R. H.: A comparison of gravity wave energy observed by VHF radar and GPS/MET over central North America, J. Geophys. Res., 105, 4685–4687, 2000.

Placke, M., Hoffmann, P., Gerding, M., Becker, E., and Rapp, M.: Testing linear gravity wave theory with simultaneous wind and temperature data from the mesosphere, J. Atmos. Solar-Terr. Phys., 93, 57–69, doi:10.1016/j.jastp.2012.11.012, 2013.

Tsuda, T., Nishida, M., Rocken, C., and Ware, R. H.: A global morphology of gravity wave activity in the stratosphere revealed by the GPS occultation data (GPS/MET), J. Geophys. Res., 105, 7257–7273, 2000.

(2) Page 5, Line 34: how do you read the new "observational filter" in de le Torre et al. (2018)? In their paper, they sort of suggests that SABER's observational window is very narrow.

de la Torre, A., Alexander, P., Schmidt, T., Llamedo, P., and Hierro, R.: On the distortions in calculated GW parameters during slanted atmospheric soundings, Atmos. Meas. Tech. Discuss., https://doi.org/10.5194/amt-2017-192, in review, 2017.

De la Torre et al. (2018) address the fact that observed altitude profiles usually are not perfectly vertical and will therefore partly sample the horizontal structure of an observed gravity wave while performing an altitude scan. This can lead to biases in the observed vertical wavelength for gravity waves of short horizontal wavelengths.

There are several reasons why this effect is very likely not important for our results:

(1) Trinh et al. (2015) included this effect in their simulation of the overall observational filter of limb sounders, and the effect was found to be small for SABER.

(2) HIRDLS and SABER momentum fluxes agree well with CRISTA momentum fluxes. CRISTA momentum fluxes, however, are unaffected by this effect because CRISTA altitude profiles were measured almost vertically (cf. Riese et al., 1999).

(3) For limb sounders the waves that pass the sensitivity function (cf. our Fig. 3) without being attenuated too much should have an aspect ratio $\lambda_z/\lambda_h$ of smaller than about 0.1, resulting in a bias of the vertical wavelength of less than $\sim$20% for SABER (cf. de la Torre et al., 2018, Fig. 7).

This information has been included in the revised manuscript on former page 10 at the end of Sect. 3.1. A more detailed discussion is given in the detailed point-by-point reply.

References:

de la Torre, A., Alexander, P., Schmidt, T., Llamedo, P., and Hierro, R.: On the distortions in calculated GW parameters during slanted atmospheric soundings, Atmos. Meas. Tech., 11, 1363–1375, https://doi.org/10.5194/amt-11-1363-2018, 2018.

Riese, M., Spang, R., Preusse, P., Ern, M., Jarisch, M., Offermann, D., and Gross-mann, K. U.: Cryogenic Infrared Spectrometers and Telescopes for the Atmosphere (CRISTA) data processing and atmospheric temperature and trace gas retrieval, J. Geophys. Res.-Atmos., 104, 16349–16367, https://doi.org/10.1029/1998JD100057, 1999.

(3) Page 6, Line 4: please consider also including Gong et al., 2015 in the reference Gong, J., J. Yue, and D. L. Wu (2015), Global survey of concentric gravity waves in AIRS images and ECMWF analysis. J. Geophys. Res. Atmos., 120, 2210–2228. doi:10.1002/2014JD022527.

Reference has been included, as recommended.

(4) Page 13, Line 21: I didn't read Gill et al., 2011, so can't comment further, but arbitrarily divide every number by two seems dangerous to me. Can you provide a rough picture like how many observed samples are 1/2 of theoretical value and what is the standard deviation?

The main problem is that, practically, it is difficult to determine "observed samples": observed altitude profiles will almost always be "contaminated" by gravity waves. For this reason, Gille et al. (2011) determined values of "measured precision" as the standard deviation calculated from of a number of consecutive altitude profiles in regions where little atmospheric variability is expected. Over a large altitude range those values are roughly a factor of two better than the theoretical ones. This indicates that the theoretical values may be high biased by this factor.

This finding is in good agreement with our Fig. 7, lower row, where we calculate the ratio:

$$R_{var} = (0.5 \cdot \text{theoretical HIRDLS precision})^2/(\text{GW temperature variance})$$

Theoretically, the maximum value that $R_{var}$ can attain is 1 (all data are noise). Values larger than 1 should not be possible. However, if we do not apply the factor of 0.5 for HIRDLS, $R_{var}$ would be a factor of 4 higher than shown in Fig. 7, lower row, and would, for example, attain values of $\sim$4 in the summer lower stratosphere (which should not be possible). For all other latitudes and altitudes, $R_{var}$ for HIRDLS would be about a factor of 4 higher than $R_{var}$ for SABER. This is also unlikely, because there is a general agreement between HIRDLS and SABER temperature variances due to gravity waves — no offset due to noise can be identified in the HIRDLS gravity wave variances shown in Fig. 5 compared to those of SABER shown in Fig. 4.

Therefore, applying the factor of 0.5 for HIRDLS is very likely justified.

In the manuscript, the text about the HIRDLS precision on former page 13 has been revised to state this more clearly.

(5) Page 15, Line 29: you may want to add "except inside the jet streams".

This information has been added, as recommended.

(6) Page 20, paragraph 1: so GRACILE only starts from z=30 km, correct? If that's the case, I strongly suggest you change Fig. 15 (I'll mention my suggestion when comes to that point).

As described on page 20, global distributions are provided at 30km and higher, however, zonal averages for HIRDLS are given also for altitudes in the range 20–30km. Therefore Fig. 15 should be kept as is in order to give users information which range of altitudes and latitudes in the tropics should be considered less reliable due to sparser sampling.

(7) Page 20, Line 13: that's where my major concern #2 comes from. If you can draw a lat-lon map you can discover whether the enhancement of sample size is source-related. Also, it would be helpful to add a bit discussion here if that's the case. Since stronger source gives you larger chance to "see" them because it's easier to find pairs.

See our reply to Reviewer # 2, Main Concern (2):

The enhanced measurement density is an effect of the satellite orbit geometry, and not an effect of the pair selection process.

(8) Page 20, Line 31: do you provide a quality flag for each value in each grid box? So user can easily make their own plots according to their needs.

No, we did not provide a quality flag.

It would also be difficult to provide a simple quality flag. As already discussed in the paper, if the ratio of temperature precision squared and gravity wave temperature variance is relatively large the data may be affected by noise. Similarly, if in a region the data are dominated by noise, we would expect high values of zonal wavenumbers. However, if in a region zonal wavenumbers are high, this does NOT automatically mean that the data are noisy — these high zonal wavenumbers could indeed be the correct values for the observed waves.

(9) Page 22, Line 14: GLIGLOSS → GRACILE?

Thank you very much! Acronym has been corrected!

(10) Fig. 10: please consider use horizontal wavelength as the colorbar so the unit and physical meaning would be consistent with Fig. 9 (or the other way around).

For convenience, additional horizontal wavelength scales have been added in Figs. 10, 16, and 17.

(11) Fig. 15: Since the sample size is not height dependent, I strongly suggest you make only line plot so you only need two plots (one for SABER with three lines as a function of latitude, and one for HIRDLS) to explain the sample size matter. The only exception is at ∼20 km for HIRDLS tropics. But since you don't provide data below 30 km because of potential cloud contamination, no need to show and discuss about that anyway.

See our reply to Minor Comment (6). Zonal averages for HIRDLS are provided at altitudes below 30km. Therefore, Fig. 15 is kept as is.

(12) Fig. 18: Instead of using dashed lines to show the natural variability, I think you can use semi-transparent grey/color areas to enclose the natural variability.

Figure has been modified, as recommended.